# RIDDEN: Data-driven inference of receptor activity from transcriptomic data

**Szilvia Barsi[1,2], Eszter Varga[2], Daniel Dimitrov[3], Julio Saez-Rodriguez[3], László Hunyady[1,2]\*, Bence Szalai[1,2¤\*]**

**1** Institute of Molecular Life Sciences, Centre of Excellence of the Hungarian Academy of Sciences, HUN-REN Research Centre for Natural Sciences, Budapest, Hungary, **2** Department of Physiology, Faculty of Medicine, Semmelweis University, Budapest, Hungary, **3** Heidelberg University, Faculty of Medicine, and Heidelberg University Hospital, Institute for Computational Biomedicine, Heidelberg, Germany

¤ Current address: Turbine Ltd., Budapest, Hungary
\* hunyady.laszlo@ttk.hu (LH); ben.szalai.cb@gmail.com (BS)

## Abstract

Intracellular signaling initiated from ligand-bound receptors plays a fundamental role in both physiological regulation and development of disease states, making receptors one of the most frequent drug targets. Systems level analysis of receptor activity can help to identify cell and disease type-specific receptor activity alterations. While several computational methods have been developed to analyze ligand-receptor interactions based on transcriptomics data, none of them focuses directly on the receptor side of these interactions. Also, most of the methods use directly the expression of ligands and receptors to infer active interaction, while co-expression of genes does not necessarily indicate functional interactions or activated state. To address these problems, we developed RIDDEN (Receptor actIvity Data Driven inferENce), a computational tool, which predicts receptor activities from the receptor-regulated gene expression profiles, and not from the expressions of ligand and receptor genes. We collected 14463 perturbation gene expression profiles for 229 different receptors. Using these data, we trained the RIDDEN model, which can effectively predict receptor activity for new bulk and single-cell transcriptomics datasets. We validated RIDDEN's performance on independent *in vitro* and *in vivo* receptor perturbation data, showing that RIDDEN's model weights correspond to known regulatory interactions between receptors and transcription factors, and that predicted receptor activities correlate with receptor and ligand expressions in *in vivo* datasets. We also show that RIDDEN can be used to identify mechanistic biomarkers in an immune checkpoint blockade-treated cancer patient cohort. RIDDEN, the largest transcriptomics-based receptor activity inference model, can be used to identify cell populations with altered receptor activity and, in turn, foster the study of cell-cell communication using transcriptomics data.

**Data availability statement:** The source code and analysis pipeline for RIDDEN are accessible at the following GitHub repository: https://github.com/basvaat/RIDDEN_analysis. The application is available at https://github.com/basvaat/RIDDEN_tool. The perturbation signatures of the receptors and the table of ligand-receptor interactions used to develop the RIDDEN model are available on Zenodo (DOI: https://doi.org/10.5281/zenodo.15127392).

**Funding:** This work was supported by the ÚNKP-22-3-II New National Excellence Program of the Ministry for Culture and Innovation from the source of National Research (SB), Development and Innovation Fund; the Hungarian National Research, Development and Innovation Fund ADVANCED 151284 (LH); National Research, Development and Innovation Office of Hungary (grant name: PharmaLab, RRF-2.3.1-21-2022-00015) (LH). BS was supported by the Premium Postdoctoral Fellowship Program of the Hungarian Academy of Sciences (460044). The funders had no role in study design, data collection and analysis, decision to publish, or preparation of the manuscript.

**Competing interests:** I have read the journal's policy and the authors of this manuscript have the following competing interests: J.S.R. reports funding from GSK, Pfizer and Sanofi and fees/honoraria from Travere Therapeutics, Stadapharm, Astex, Pfizer, Grunenthal and Owkin. B.S. is a full time employee of Turbine Ltd. and reports fees from TheraMe! AG.

## Author summary

Receptors play a crucial role in intercellular communication, thereby influencing essential physiological processes. By identifying which receptors are active and initiating signaling within cells we can gain insights into the processes triggered by this communication. Changes in gene expression patterns regulated by these receptors allow us to infer their activity, providing more insight than measuring the receptor or ligand gene expression levels, which often fail to accurately reflect the actual protein activity within the cells. To address this, we developed RIDDEN (Receptor actIvity Data Driven inferENce), a tool for predicting receptor activity by summarizing gene expression profiles regulated by receptors into interpretable activity profiles. We evaluated RIDDEN's performance and found that it reliably captures receptor activity and its biological implications. Thus, RIDDEN enhances our understanding of cellular processes related to communication between cells and helps identify the sources of signaling that can lead to various cellular phenotypes.

## 1. Introduction

Ligands and receptors are considered the key drivers of cell-cell communication (CCC), which process is essential for physiological functions. The ligand produced by the "sender-cell" reaches the "receiver-cell", where it binds to its receptor and changes its activity. This binding initiates downstream signaling from the receptor, and, as a result, it alters the receiver cell's state. This process is vital for cells to respond to their environment, which regulates essential processes, such as maintaining homeostasis, cell growth, differentiation or immune interactions. Dysregulation of the receptor activation, caused by altered ligand binding, mutation or overexpression of the receptor, can lead to various diseases, including changes in insulin, neurotransmitter, G protein-coupled or overexpression/overactivation of growth factor receptors leading to insulin resistance [1], neurological disorders [2], endocrine diseases [3] or cancer development [4], respectively.

Experimental investigation of ligand-receptor interactions on the systems biology scale remains challenging due to the complexity of cellular interactions [5,6]. The limited scope of studying a few isolated cells experimentally restricts the ability to obtain comprehensive information on communication [7]. A large number of computational methods have emerged to discover and analyze these interactions in the last decade [6]. Most methods use prior-knowledge-based lists of receptor-ligand interactions coupled with statistical methods to identify sender-receiver cell pairs with significant expression of corresponding ligand-receptor pairs [8–13]. Several new methods [14–16] focus on the ligand-induced expression changes of "receiver-cell" to infer the activity of ligand-induced signaling, however no current method is able to infer receptor activity directly.

Computational methods predominantly rely on transcriptomic data because it is broadly available, and RNA sequencing is a high-throughput and cost-effective

approach [17]. A limitation of transcriptomics data is that gene expression cannot be directly translated to protein levels due to regulatory mechanisms, splicing, or post-translational modifications [5,18]. However, if gene expression changes are considered a "footprint" (consequence) of altered protein activity, activity can be computationally inferred from gene expression changes. Methods using these ideas are generally called "footprint-based" tools [19]. These methods require information about the genes regulated by the proteins of interest. While these regulatory interactions are well-characterised in the case of transcription factors [20], in the case of other signaling proteins, like receptors, less information is available. Perturbation gene expression signatures, where gene expression is measured after the perturbation of a given protein, can help to identify such protein-gene regulatory interactions [20,21].

Building on this principle, a recent study [14] introduced a method (CytoSig), that predicts cytokine signaling activities from transcriptomic profiles. It is based on a collection of cytokine stimulation signatures accessible in public databases, like the Gene Expression Omnibus [22]. While CytoSig effectively predicts the activity changes for 43 cytokines, the authors describe that lack of data is a limitation of the research field. Some cytokines were trained on a low number of expression profiles. In addition, these experiments were not conducted based on a standardised method, and the use of different platforms for sequencing led to high variability between samples.

We developed RIDDEN (Receptor actIvity Data Driven inferENce), a statistical model that infers receptor activities from transcriptomic profiles to overcome the limitation of the low number of high-quality signatures in public databases and to extend the number of predictable signaling molecules. To enable the inference of receptor activities, we combined the advantages of the prior knowledge of ligand-receptor interactions from the OmniPath resource [23,24] and a large collection of uniformly conducted and processed ligand and receptor perturbation gene expression experiments from The Library of Integrated Network-Based Cellular Signatures (LINCS), which is the most extensive collection of gene expression profiles of perturbations applied in a wide range of time points, doses and several cell lines [25]. We developed a model to predict the activity of 229 receptors and demonstrated its performance by predicting cytokine signaling activities in comparison with the state-of-the-art method, the CytoSig [14]. The RIDDEN resource and console application are accessible at https://github.com/basvaat/RIDDEN_tool. We show how the collected resource correlates with fundamental biological processes and a case study where receptor activity can be used as a biomarker of patient survival in immunotherapy.

## 2. Results

### 2.1. Establishment of the RIDDEN model for receptor activity inference

As the first step of constructing the statistical model that infers receptor activity from transcriptomic profiles a comprehensive dataset containing receptor and ligand perturbation data was required. We obtained curated ligand-receptor interactions from the OmniPath database [23,24]. Based on this prior knowledge, we collected the chemical (drug treatment, ligand stimulation) and genomic (knock-out, knock-down, overexpression) perturbation profiles of all the receptors and ligands available in the LINCS L1000 database [25]. We used level 5, log-normalized, standardised gene expression profiles in our study. In total, our database consisted of 38989 consensus transcriptional profiles for 599 receptors and ligands in 5 different perturbation types, and these data serve as the basis of the statistical model.

For training RIDDEN, we fitted linear regression models on the receptor perturbation gene expression data, using the known receptor perturbations (+1 activated, -1 inhibited, 0 not perturbed) as input [26,27]. Linear models, despite their simplicity, remain among the most widely used model types for gene expression signature-based studies [14,26,28]. We fitted the models for each receptor-regulated gene pair and used the coefficients of the linear regression to define the RIDDEN model (Fig 1A). We use a permutation-based approach to estimate receptor activities in new samples (Methods, Fig 1B).

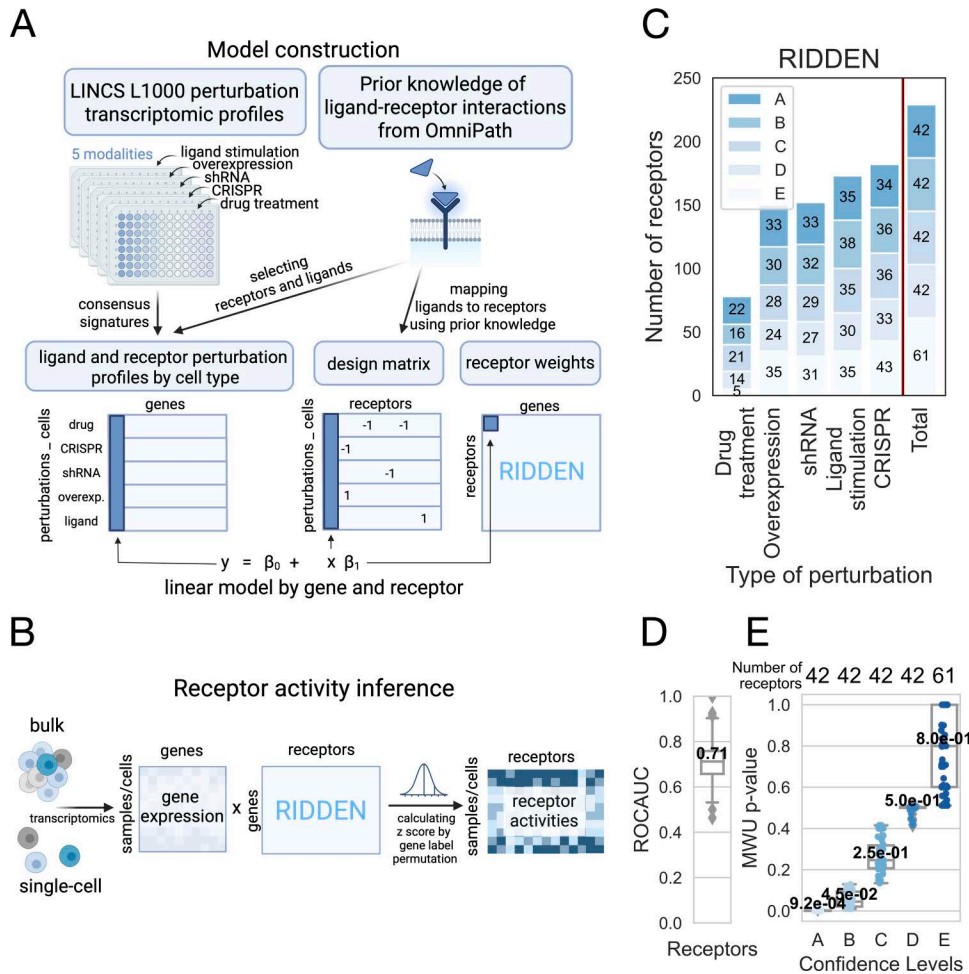

**Fig 1. RIDDEN Construction of the model and workflow.** (A) RIDDEN model construction. The perturbation profiles are collected from the LINCS L1000 database, and the ligand-receptor interactions are collected from OmniPath. The receptor and ligand perturbation signatures are then filtered from LINCS, and consensus gene expression signatures are calculated. Next, the ligands are mapped to corresponding receptors, known to be interacting according to prior knowledge, and linear models are fitted on the ligand-receptor perturbation profiles using the known receptor perturbations (+1 activated, -1 inhibited, 0 not perturbed) as input to create the RIDDEN. Created in BioRender. (B) Inference of receptor activities using RIDDEN. From bulk and single-cell transcriptomics, RIDDEN estimates receptor activities using dot products and computes z-scores using permutations of gene labels. Created in BioRender. (C) Number of receptors in RIDDEN. Barplots show receptor counts by perturbation types. Each bar shows the number of receptors corresponding to each confidence level. (D) The cross-validation performance was assessed using ROC AUC. The boxplot shows the median ROC AUC of 229 receptors, the first and third quartiles. The minimum and maximum values are shown as whiskers. (E) Receptor confidence levels. The boxplot shows the distribution of Mann-Whitney U (MWU) p-values for receptors across different confidence levels. The p-values were calculated using the mean of the splits applied during the evaluation process. The x-axis represents the confidence levels (A-E), and on the top, the number of receptors in each confidence group is shown. The median p-values are depicted in black, while the first and third quartiles are represented by the box, and the minimum and maximum values are shown as whiskers.

To ensure that our statistical model only includes receptors demonstrating predictive capability, we kept only signatures passing certain criteria. For quality filtering, we first constructed 5 models for the 5 perturbation types (ligand, compound, CRISPR, etc.) separately, and we performed cross-validation to filter for receptors whose activity can be predicted at least in one other modality (Methods). Finally, the RIDDEN summarizes 14463 high-dimensional gene expression profiles for 229 receptors into differential receptor signatures that can be leveraged to infer receptor activities from bulk and single-cell transcriptomics. The distribution of expression values for each gene across all receptor

perturbations is shown in S1 Fig. Next, we conducted an additional cross-validation and calculated the predictive performance for all 229 receptors, for which we used the complete set of their perturbation signatures from every modality (Fig 1C, Methods). This model (Fig 1C, labeled with Total) was then applied for subsequent analyses. The cross-validation resulted in a median ROC AUC score of 0.71. (Fig 1D). Based on the cross-validation performance (Fig 1E, Methods), we further classified receptor signatures into five confidence levels (A-E, where A confidence level receptors had the best cross-validation performance). For further evaluation, we use the RIDDEN model fitted on all perturbation types.

## 2.2. Model benchmark

To assess the prediction performance of RIDDEN, we used a dataset of bulk cytokine perturbation transcriptomic profiles collected by the authors of CytoSig, and for further evaluation, we used the most recent Immune Dictionary's single-cell transcriptomic profiles in response to many different cytokines as ground truth. RIDDEN was used to predict the cytokine receptor activities of the samples, and we then compared its predictive performance with that of the CytoSig model in predicting cytokine signaling activities. We argued that if the model effectively predicts receptor activity or cytokine signaling activity from gene expression data, it can identify the perturbed receptor or cytokine. This was evaluated using the ROC AUC metric (Fig 2A).

First, to compare RIDDEN with CytoSig, we performed a cross-evaluation by testing each model on the dataset that was used to train the other model. Although RIDDEN was trained on LINCS landmark genes, we evaluated the prediction of CytoSig using both the landmark and the inferred genes in the LINCS signatures. The full RIDDEN yielded 0.61, while the CytoSig 0.59 median ROC AUC. Moreover, the RIDDEN receptor models with confidence levels A-E reached 0.68, 0.64, 0.56, 0.51 and 0.52, respectively (Fig 2B).

Next, we used the single-cell transcriptomic profiles of the Immune Dictionary [15] to determine if the RIDDEN and CytoSig methods could predict which cytokine was perturbed in the sample. We analyzed the cytokines and their receptors that were common in all three datasets. Here, we saw that despite not explicitly modelling ligand perturbations, the RIDDEN had a comparable ROC AUC score of 0.64 with CytoSig (0.67). We then assessed the prediction of cytokine receptors using the entire Immune Dictionary dataset that included all identifiable cytokines. The ROC AUC scores for receptor categories A-E are 0.84, 0.55, 0.61, 0.56, and 0.44, respectively (Fig 2C).

To compare the performance of RIDDEN to other ligand activity prediction methods, we also used NicheNet (NN) to predict ligand activity of Immune Dictionary cytokine perturbation data. The ligand activity predictions of the NN resulted in a median ROC AUC of 0.54, while evaluating 64 overlapping cytokines with Immune Dictionary (S2 Fig). RIDDEN achieved higher ROC AUC score on the three dataset overlap (Fig 2C), which included 28 cytokine receptors and their 58 interacting cytokines, 46 of which also overlapped with NN.

Taken together, RIDDEN achieved predictive performance similar to CytoSig in both the cross-validation setup and the independent perturbation dataset. Notably, for the receptors with the highest confidence, RIDDEN outperformed CytoSig (Fig 2B, C).

## 2.3. Model evaluation and correlation with biology

Besides benchmarking predictive performance, we also analyzed RIDDEN from a biological perspective. To achieve this, we examined whether receptors that share similar signaling pathways are clustered based on model weights. We also analyzed the relationship of the receptor with transcription factor (TF) activity. Finally, we calculated the predicted receptor activity correlation with baseline receptor and ligand expression of cell lines and patient samples.

First, we investigated the similarity of receptors in the model by hierarchical clustering of receptor weight vectors. We can identify receptor family members clustered together on the dendrogram due to similar downstream signaling (Fig 3A). We performed Gene Ontology biological process [29] enrichment analysis on gene weights of the RIDDEN receptors

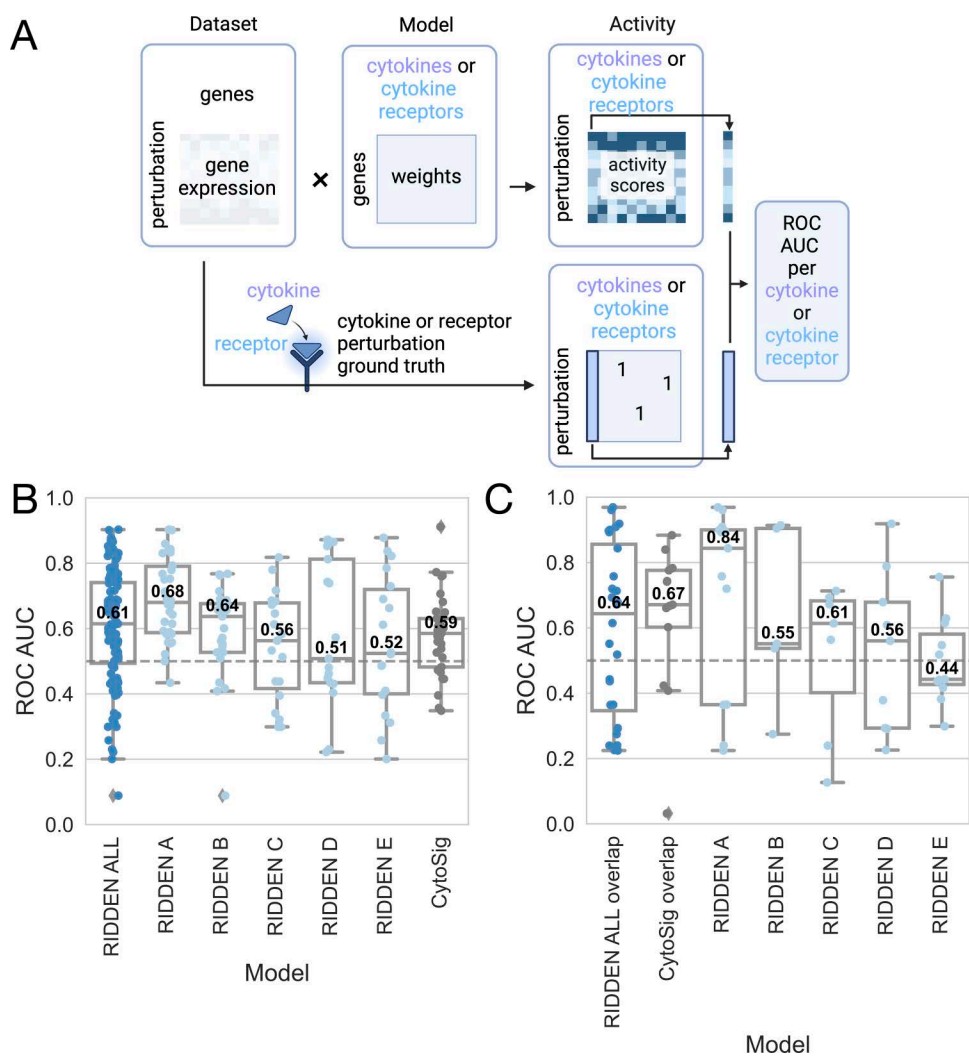

**Fig 2. Model benchmark.** (A) The receptor activities are inferred from cytokine stimulation and cytokine receptor perturbation gene expression using the CytoSig and RIDDEN models. The ground truth describing the perturbed cytokine or receptor in the sample is used to calculate the ROC AUC per cytokine in the case of CytoSig prediction or cytokine receptor in the case of RIDDEN prediction. Created in BioRender. (B) Comparison of the predictive performance of the RIDDEN and the CytoSig model. The boxplot shows the distribution of ROC AUC values derived from predictions with the CytoSig and RIDDEN models on the other model's training dataset. The RIDDEN model matrix was split into subsets based on confidence scores, and these models were tested against CytoSig. In the case of the CytoSig model, the evaluation dataset is the LINCS cytokine perturbation signatures consisting of landmark genes + inferred genes. The median ROC AUC values (central line and white numbers), first and third quartile (box), minimum and maximum non-outlier values (whiskers) and outliers (diamonds) are shown on the boxplot. (C) Benchmarking of the RIDDEN and the CytoSig models on the Immune Dictionary dataset. The boxplot shows the distribution of ROC AUC values. The first two boxes (RIDDEN ALL and CytoSig ALL overlap) represent the evaluation of models on the overlapping cytokines and their receptors of the three resources. The cytokines of Immune Dictionary were mapped to receptors of RIDDEN, thus resulting in a different number of data points in this comparison. The following boxes show the ROC AUC values of RIDDEN models predicting the activities of all overlapping Immune Dictionary cytokine receptors. These models contain receptors with different confidence scores (A-E). The boxplot features are as described in B.

with distinct functions, to assess whether the model captures biologically meaningful patterns and to validate that genes with high absolute weights reflect relevant downstream processes of each receptor. As some representative examples, the top enriched terms aligned with known receptor functions, such as CXCR4 (C-X-C chemokine receptor type 4, a G protein-coupled chemokine receptor), which is associated with lymphocyte/leukocyte homeostasis [30]; IFNGR1

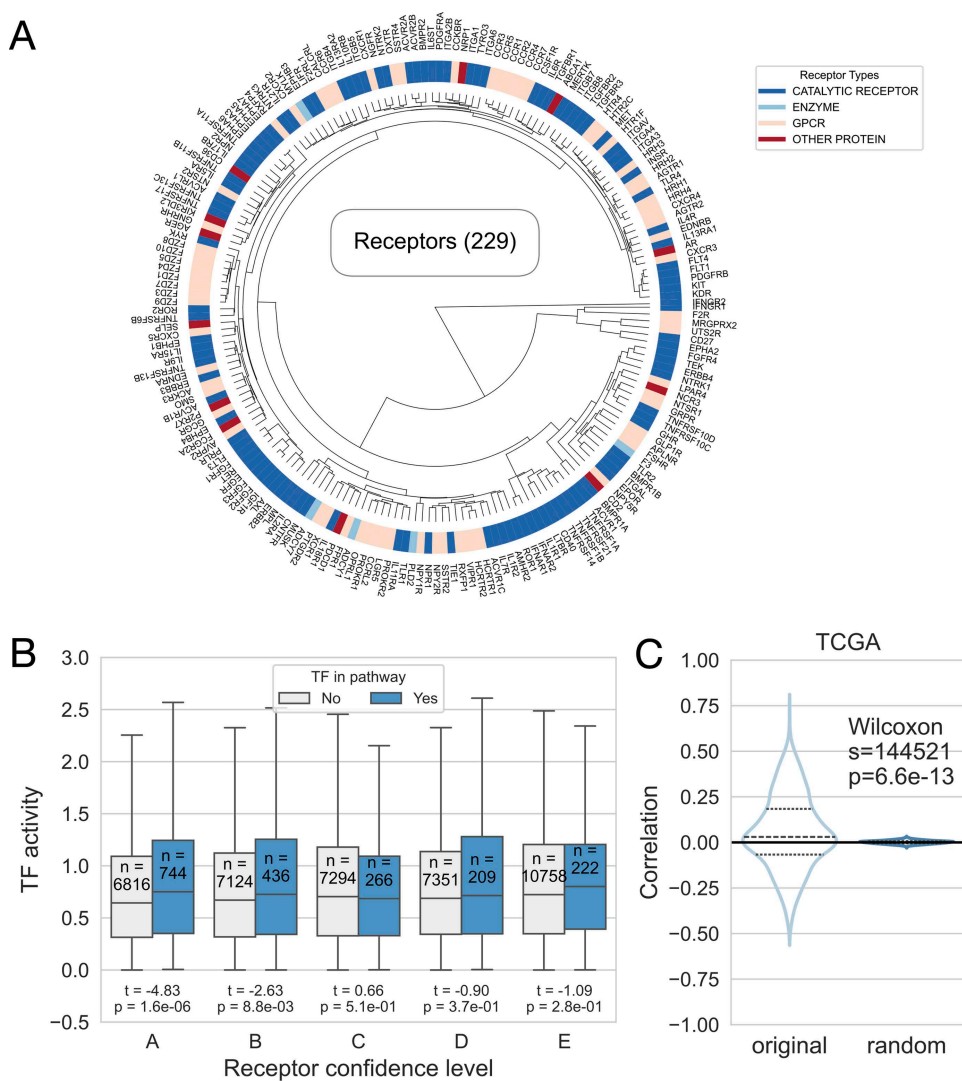

**Fig 3. Evaluation of the model.** (A) Hierarchical clustering of the receptor weights of the RIDDEN model. The colours indicate different receptor families: catalytic receptors, enzymes, G-protein coupled receptors and other receptors, such as transporters or ion channels, from red to blue. (B) Relationship between TF activities and receptors in the same pathways. The pairs of boxes show the calculated TF activities from the RIDDEN receptor weight vectors when the TF and the receptor have a shared KEGG pathway (light box) or not (dark box) at different confidence levels. Each pair of boxes represents the receptors with different confidence levels. The number of pathways and median TF activities, first and third quartile (box), and minimum and maximum non-outlier values (whiskers) are shown on the boxplot. (C) Correlation analysis of the TCGA samples receptor * ligand TPM values and their receptor activities. The distribution of correlation values in the dataset (original - left) and the random distribution (1000 gene label permutation - right) are represented in a violin plot. A dashed line indicates the quartiles of the correlation values.

(interferon gamma receptor 1, a type II cytokine receptor) has a role in antigen processing and presentation [31]; IL6R (interleukin 6 receptor) os crucial for regulating immune responses as a cytokine receptor [32], and also plays a role in regulating autophagy [33]; TNFRSF1B (tumor necrosis factor receptor superfamily member 1B, a TNF receptor) is linked to T cell activation in immune response [34,35]; and TGFBR1 (transforming growth factor beta receptor 1, a TGF-beta receptor) is essential for all phases of wound healing [36,37], and is directly involved in hemostasis [38] (S1 Table). While enriched terms reflect receptor biology, complete overlap is not expected because methods like RIDDEN capture functional changes based on downstream effects rather than relying on predefined gene set enrichment.

RIDDEN infers receptor activity by capturing the gene expression changes induced by receptor perturbation. However, the gene expression does not necessarily reflect the activity of the proteins, therefore, the receptor-induced gene expression signature is expected to correlate more with the activity of downstream effectors, such as TFs, that directly regulate target genes in response to receptor signaling, rather than with the gene expression levels of those effectors themselves.

Active receptors initiate signaling cascades that modulate the activity of downstream TFs. Thus, we anticipated that the absolute activity of TFs, which are downstream to the active receptors is increased compared to TFs, which are not affected in the signaling pathway. Therefore, we estimated transcription factor activities of receptor model weights using decoupleR [28] with TF regulons from DoRothEA [20]. The RIDDEN model's weights represent the receptor-induced gene expression signature. We compared the activities of transcription factors included or not in the same KEGG signaling pathways [39] as the receptors. We found that TF-receptor pairs sharing KEGG pathways had significantly higher activity than those that do not share KEGG pathways (t-test p-value = 1.6x10-5). We also saw significant differences in receptors with the highest confidence scores, p-values of $1.6x10^{-6}$ for A-level receptors and $8.8x10^{-3}$ for B-level receptors. (Fig 3B). Additionally, we correlated receptor activity with both absolute TF activity and TF gene expression levels calculated from the CytoSig dataset. We found that the difference in correlation distribution between TF–receptor pairs within the same KEGG pathway versus unrelated pairs was substantially greater when using TF activity (t-test p-value = 9.3x10-54) compared to TF gene expression (t-test p-value = 1.5x10-22) (S3 Fig).

To demonstrate the significance of predicting receptor activities, we performed a correlation analysis based on the assumption that, if the ligand and its receptor are upregulated, their increased accessibility indicates the potential for receptor activation. For this analysis, we used patient tumor samples from The Cancer Genome Atlas (TCGA) [40], where cancer cells are bulk sequenced together with their microenvironment thus, paracrine signaling can be assessed. We also performed our analysis on the more homogenous cancer cell line data from the Cancer Cell Line Encyclopedia (CCLE) [41]. We investigated the distribution of Pearson's correlations between receptor activity and the product of receptor and its ligands expression. Although the average correlation was relatively low, the distribution significantly differs from the random distribution with p-value = 6.6x10-13 and 1.7x10-2 for TCGA (Fig 3C) and CCLE (S4 Fig) data, respectively.

To summarize, these findings suggest that the integrated receptor perturbation signatures capture biological processes and that the model can be leveraged for receptor activity prediction.

## 2.4. RIDDEN identifies biomarkers for cancer therapy response

After evaluating RIDDEN's prediction performance and biological validity, we analyzed its potential to predict patient response to cancer therapy. Here, we used an immune checkpoint therapy dataset, as immune-oncology therapies rely on disturbing the interactions between cancer and immune cells, thus cell-cell communication methods, in this case, can be especially relevant. Immune checkpoint blockade (ICB) therapies have emerged as a promising advancement in cancer treatment. They target immune checkpoint molecules, which are pairs of ligands and receptors responsible for regulating immune responses, such as the PD-1 and PD-L1. PD-1 is expressed on the surface of various immune cells, and activated PD-1 inhibits the anti-cancer activity of these cells. PD-L1 is prominently expressed on the surface of tumor cells and activates PD-1, thus facilitating the immune escape of cancer cells. While ICB is effective in a group of patients, it is important to identify biomarkers to determine which patients can benefit from the therapy [42].

We analyzed the data from a study, where clear renal cell carcinoma patients were treated with either nivolumab, a PD-1 blockade therapy, or everolimus, an mTOR inhibitor [43]. We focused on the pretreatment samples, where gene expression was measured less than one year prior to the initiation of the therapy. We analyzed the associations between PD-1 receptor expression/ PD-L1 ligand expression/ PD-1 receptor activity and patient overall survival (OS). There was no association between OS and PD-1 or PD-L1 gene expression in the nivolumab-treated patient cohort (log-rank test p-value = 0.465, 0.318, respectively) (Fig 4A). Although ligand or receptor mRNA levels were not predictive of patient survival, the RIDDEN-estimated PD-1 receptor activity was associated with the survival of nivolumab-treated

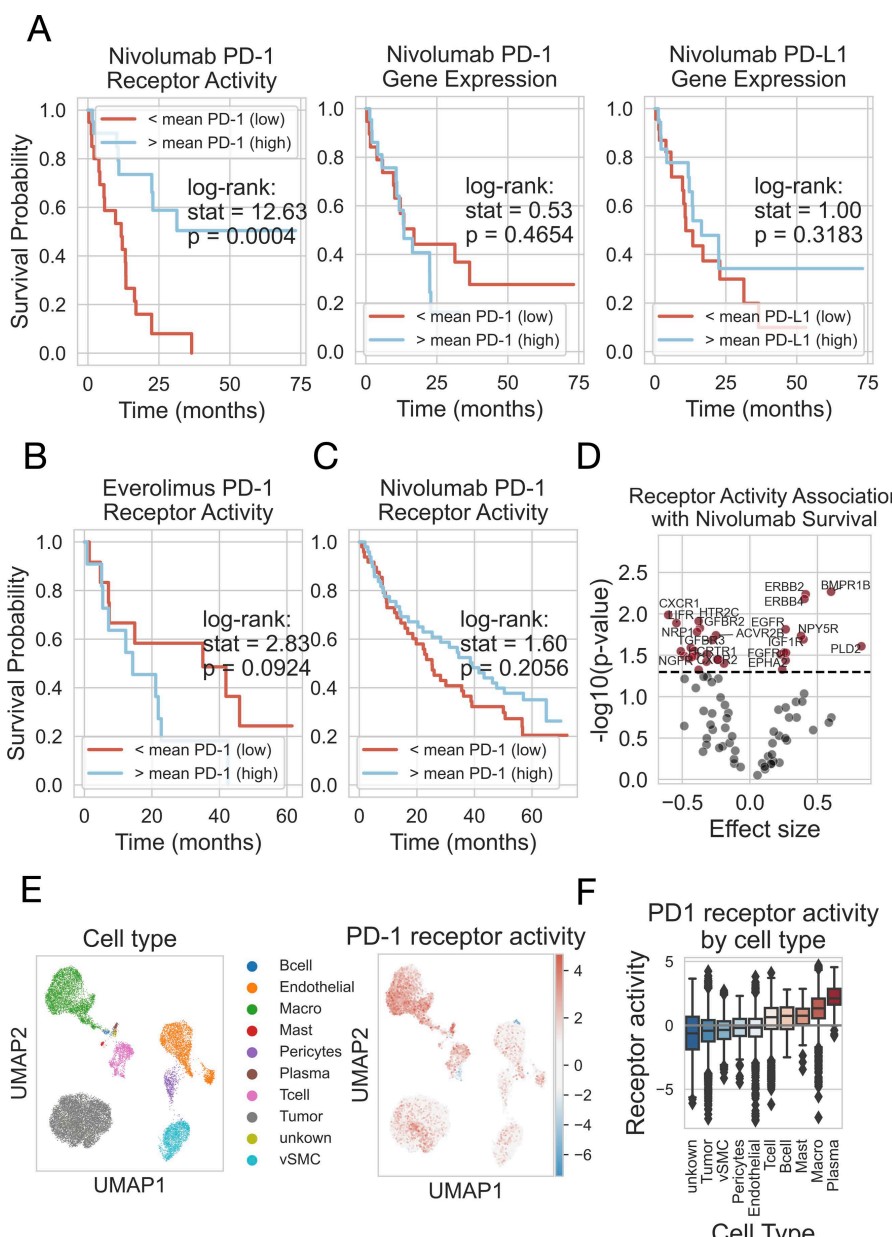

**Fig 4. PD-1 expression and activity in tumor samples.** (A) PD-1 receptor activity is associated with the survival of patients with renal cell carcinoma, whereas PD-1 and PD-L1 gene expression are not associated. Kaplan-Meier plots represent the overall survival in months (x-axis) and the survival probability (y-axis) of samples with high or low activity of PD-1 receptor (left), expression of PD-1 receptor (middle) and expression of PD-L1 ligand (right), where the mean value threshold is used. The result of the log-rank test is shown. The red line indicates a low; the blue indicates a high expression/activity. (B) The activity of the PD-1 receptor was not associated with patient survival in the case of patients treated with everolimus, an mTOR inhibitor. (C) The activity of the PD-1 receptor is not associated with survival in samples that were taken more than a year before treatment with nivolumab, a PD-1 inhibitor. (D) Association of receptor activity with patient survival in response to nivolumab. The volcano plot shows the Cox regression coefficients (effect size) (x-axis) and the $-\log_{10}$(p-values) (y-axis) for the receptors with A and B confidence (dots). The black dashed line indicates the p-value < 0.05 significance level. Receptors with a p-value < 0.03 are shown on the plot. (E) A 2-dimensional representation of the high-dimensional single-cell transcriptomic profile of RCC patients. The UMAP shows the different cell type clusters with different colors (left) and the PD-1 receptor activity of the single cells (right), where high activity is marked by red and low activity is marked by blue. (F) The PD-1 receptor activity of different cell types in samples from 7 patients with renal cell carcinoma. The x-axis represents cell types, while the y-axis represents PD-1 receptor activity. The median activity values (central line), first and third quartile (box), minimum and maximum non-outlier values (whiskers) and outliers (diamonds) are shown on the boxplot.

(p-value = 4*10-4), but not with the everolimus-treated patients (p-value = 9.2*10-2) (Fig 4A, B). The effect of PD-1 receptor activity on the overall hazard in the nivolumab-treated patient cohort was found to be negative (Cox regression β = -0.36, p-value = 0.012) (Table 1). This means that higher PD-1 receptor activity was associated with a decrease in the hazard rate, leading to a longer survival time in patients treated with nivolumab. However, this effect was not significant in patients treated with everolimus, and the association was in the opposite direction (β = 0.33, p-value = 0.106) (Table 1). As nivolumab inhibits PD-L1/ PD-1 interaction, the observed increased effectiveness of the drug in high PD-1 activity corresponds to its mechanism of action. The gene expression of the ligand or the receptor showed no association with survival in everolimus-treated patients (Table 1). Importantly, in the patient cohort, where the samples were taken more than one year before the treatment, the PD-1 receptor activity did not show an association with patient survival in any of the examined cases (Fig 4C). In the samples of nivolumab-treated patients, chemokine receptors (CXCR1, CXCR2), serotonin receptor HTR2C, TGF-β receptors (TGFBR2, TGFBR3), or LIFR exhibited significant negative effects, and BMPR1B, members of the ERBB receptor family, and EGFR had the most significant positive effect on the overall hazard in Cox regression analysis. (Fig 4D and S2 Table). The Cox regression results for gene expression of receptor genes among nivolumab-treated patients and receptor activities in the everolimus-treated patient cohort are available in S3 and S4 Tables, respectively.

Furthermore, we investigated whether RIDDEN receptor activity signatures reflect the potential effects of PD-1 activation in different cell types. For this purpose, we analyzed a dataset containing 7 renal cell carcinoma patients' single-cell gene expression profiles [44]. Each cell shown in the UMAP is clustered by expression after batch correction (Fig 4E). We calculated the PD-1 receptor activities in these cells and showed that the immune cells have high activity of PD-1, while no activity was observed on tumor cells, endothelial, perivascular and smooth muscle cells. (Fig 4E and F).

## 3. Discussion

RIDDEN (Receptor Activity Data-Driven Inference) is a computational tool developed to infer receptor activities by summarizing thousands of ligand and receptor perturbation gene expression profiles into interpretable receptor activity states. Receptors play a crucial role in both health and disease by initiating and mediating signaling during cell-to-cell communication. By providing insights into the initiators of signaling within cells, RIDDEN enhances our understanding of cellular processes and supports research hypotheses, including assessing the effects of drugs on receptor activity or identifying sources of the abnormal signaling associated with disease phenotypes.

RIDDEN combines an extensive collection of receptor and ligand perturbation transcriptomic profiles [25] and prior knowledge of ligand-receptor interactions [23,24]. The collected dataset enables us to summarize key changes upon receptor activity in different samples. Current state-of-the-art methods in predicting cell communication are typically based on the prior knowledge networks of ligands and receptors. While these methods can give valuable insight into the potential receptor-ligand interactions and communicating cell types, RIDDEN directly infers the receptor level effect of these

**Table 1. The effect of PD-1 receptor activity and PD-1 or PD-L1 gene expression on the overall survival of patients diagnosed with renal cell carcinoma determined by Cox regression analysis.**

| Gene name | Type | Treatment | Effect size | p-value |
|---|---|---|---|---|
| PD-1 | gene expression | nivolumab | −0.0513 | 0.5849 |
| PD-L1 | gene expression | nivolumab | −0.0694 | 0.6299 |
| PD-1 | receptor activity | nivolumab | −0.3622 | 0.0124 |
| PD-1 | gene expression | everolimus | −0.0256 | 0.8604 |
| PD-L1 | gene expression | everolimus | 0.1422 | 0.4779 |
| PD-1 | receptor activity | everolimus | 0.3320 | 0.1061 |

interactions on the receiver cell. It contains high-quality receptors that are validated with models constructed from other modalities. In addition, as a footprint-based method [45], it derives more biological insights than gene set-based methods.

While the RIDDEN model is based on high-throughput measurements, limitations need to be noted. The perturbation profiles consist of 978 landmark genes and do not use the inferred set of more than 10,000 genes [25]. Although whole transcriptome measurements could help distinguish signals between receptors with similar downstream signaling, they are not available at this scale. However, a promising approach has been introduced, which measures single-cell transcriptomic profiles of immune cell types on cytokine perturbation [15], yet there is a need for data on other ligands and receptors that are crucial in intercellular communication.

The increasing number of computational models developed to study the CCC brings attention to the importance of benchmarking and evaluating such models. In computational biology, one of the major challenges is the common lack of ground truth to evaluate predictive models [46]. In terms of receptor activities, cytokine perturbation gene expression data collected by CytoSig [14] or NicheNet [10] can be a reasonable basis with regard to cytokine receptors. The NicheNet model is trained on a small coverage of public experiments, and more than half of its data overlaps with CytoSig. Additionally, the data available in public databases is often specific to a particular experimental design and only measures a pair of cell lines or conditions. The existing studies mainly evaluate the model performance using collected experimental data [9] and bulk or single-cell tissue measurements [10,14,47]. This may change with new studies, such as the Immune Dictionary [15], which relies on *in vivo* data. Furthermore, comparing RIDDEN with classical CCC methods on perturbation datasets would not yield meaningful insights, as these methods rely on endogenous ligand expression and are unable to capture the externally introduced ligands, making them unsuitable for estimating interactions with receptors in perturbation experiments.

The CytoSig model is suitable for benchmarking against our model as it predicts the activity of ligand-stimulated signaling using cytokine perturbation bulk datasets, similar to how RIDDEN employs cytokine receptor perturbation datasets. RIDDEN performed comparably to CytoSig in cytokine signaling or cytokine receptor activity predictions, moreover, RIDDEN provides confidence levels for receptors. In the evaluation of the top-performing receptors, the model outperformed CytoSig, demonstrating its robust performance in predicting cytokine receptor activity. RIDDEN has an advantage over this model due to its broader coverage of perturbation signatures within the same experimental design, encompassing a greater number of receptor and ligand perturbations. This allows the model to capture the core signaling changes induced by receptor activity that occur similarly across different cells.

We compared our model to the published CytoSig model and evaluated its performance using *in vivo* ligand perturbation dataset from the Immune Dictionary. *In vivo* experiments model more complex responses to perturbations as they capture cellular responses within their microenvironment within a tissue. RIDDEN's ability to predict the receptor activities induced by ligand stimulation in a complex environment suggests that by learning from *in vitro* data, RIDDEN can capture the fundamental responses of receptor activation. RIDDEN predicted which receptor-induced signaling was modulated by the immune ligand with performance comparable to CytoSig.

Besides benchmarking the prediction performance, we also analyzed RIDDEN's performance from a biological perspective. First, we show that in the RIDDEN model receptor family members have highly similar signatures consistent with biological expectations because they may form heterodimers during activation, bind the same ligands or have shared effectors. Then, we used two large transcriptomic datasets, TCGA and CCLE, for evaluation and testing of further fundamental biological concepts, such as the correlation of receptor-ligand expression with receptor activity. Another strategy is using pathway gene sets to validate the connection between the activity of transcription factors and the activated receptors that could potentially affect them. Although we observed low correlation in these comparative analyses, we saw that our model performed better than random. We anticipate such a low correlation given the expected disagreement between expression and activity [48]. This observation highlights the importance of predicting activities. The results of these analyses suggest that we can provide a good approximation of these processes.

In addition to physiological processes, the model's applicability extends to examining cell lines carrying a disease phenotype or patient samples. We present a case where the receptor plays a vital role in cancer therapy. Immune checkpoint inhibitors, especially the PD-1 receptor inhibitor, are widely used in cancer therapy [49]. One of the criteria for patients receiving anti-PD-1 treatment is whether they carry high PD-L1 expression measured by immunohistochemical staining of the tumor and immune cells [50]. This suggests if inhibiting the receptor can be effective in the patient. We cannot determine whether the patient will benefit from the therapy based on the gene expression of the PD-L1 ligand or even the PD-1 receptor. However, estimating the activity of receptor proteins can serve as a reliable predictor for the effectiveness of therapy. We have found that the patient's overall survival on anti-PD-1 therapy is associated with increased PD-1 receptor activity before the treatment. Additionally, we have observed associations between patient survival and the activity of several other receptors already connected to immune oncology. These include chemokine receptors that play a crucial role in communication between cells in the tumor microenvironment (TME) [51,52], serotonin receptors that have been reported to impact cancer progression by influencing immunological processes [53], such as promoting immunosuppressive M2-like macrophage polarization [54], or and TGFB receptors whose signaling promotes tumor immune evasion in TME [55], These receptors are activated in the TME where the PD-1 signaling can be enhanced [56].

Importantly, PD-1 activity is associated with increased survival only in the nivolumab-treated cohort and not in the everolimus-treated patients, underlying the specificity of the prediction method.

When investigating the association between mutations of patient samples and patient survival, the time of sample collection is less crucial than examining changes in gene expression or receptor activity and their association with patient survival. In this regard, the timing of sample collection can have a significant impact, as various factors, regulatory mechanisms, and cellular processes can influence gene expression. The activity of the PD-1 receptor in samples taken less than one year before anti-PD-1 therapy initiation is associated with the overall survival of patients, but this association is not observed in samples taken more than 1 year before therapy initiation.

In addition, we have also shown that we can precisely detect the activity of the receptor in the cell types that may be present, such as T cells [57] or tumor-associated macrophages [58]. On the other hand, we do not predict receptor activity in tumor cells due to the lack of a signature induced by the active receptor.

In summary, the RIDDEN method is a reliable and easy-to-use tool for inferring receptor activities across biological contexts, which can be used to obtain a comprehensive overview of active and inactive receptors from transcriptomics data.

## 4. Methods

### 4.1. Collection of ligand and receptor perturbation signatures

We queried the OmniPath database [23,24] for curated ligand-receptor interactions using the OmniPath R package to obtain the most reliable collection of possible interactions. From there, we retained all of the receptor and ligand perturbation signatures from the LINCS database from 5 modalities: the genetic (shRNA, CRISPR, overexpression) and the chemical (ligand stimulation, drug treatment) perturbations. For model construction, we used the level 5 LINCS L1000 signatures and landmark gene set. To calculate consensus signatures for all cell and perturbation pairs, we aggregated the perturbation signatures and computed moderated-Z weighted averages for every perturbation in each cell line, following the method outlined in the LINCS publication [25,59].

Finally, the final dataset comprises 14463 perturbation profiles, where each signature corresponds to a receptor perturbation transcriptomic profile in a cell line, referred to as a sample. This dataset includes 229 different perturbed receptors, 228 distinct cell lines, and 747 unique perturbations derived from the different data modalities.

### 4.2. Construction of the model

After collecting and calculating the consensus gene expression profiles of all possible ligand and receptor perturbations, we constructed linear regression models for the expression of all genes and receptor perturbations.

We used ordinary least squares (OLS) regression models (statsmodels Python package) to estimate the relationship between receptor perturbation and the expression of a gene. In the following linear model equation: $g_i = \beta_0 + \beta_j\, r_j + \epsilon$, where the predictor $r_j$ is a vector that represents samples (cell-receptor perturbation pairs), where each value denotes the perturbation of receptor $j$ in the sample. The values are encoded as follows: 1 denotes stimulation, -1 denotes inhibition, and 0 is no perturbation. If a ligand is perturbed, curated ligand-receptor interactions are used to translate this ligand into corresponding receptor perturbations. The response variable $g_i$ is a vector of gene expression values, with each value representing the expression level of a specific gene in each sample. The $\beta_0$ denotes the intercept term, $\beta_j$ is the coefficient for the receptor $j$ (the parameter of interest), and $\epsilon$ is the error term.

Each linear model generated a coefficient ($\beta_j$) for each receptor-gene pair, indicating the strength and direction of the receptor perturbation's influence on gene expression. The coefficients from each regression model were organized into a receptor-gene parameter matrix, reflecting the relationship between receptor perturbations and gene expression under different conditions (perturbation type, direction, and cell line). This parameter matrix summarizes the high-dimensional gene expression profile associated with variations in receptor activities.

### 4.3. Inference of receptor activities from gene expression

To estimate the receptor activity, we calculate the dot product of the gene expression profile of the samples and the RIDDEN matrix (described in 4.2.). This method aligns with a previously described approach, according to pathway activities that can be inferred from the sample's transcriptional profiles [26]. An additional step involves generating a background distribution of receptor activities by performing 1000 permutations of gene labels. Subsequently, we calculate the standardised score (z-score) of the value relative to the background.

### 4.4. Improving the quality by signature filtering based on cross-validation

We aimed to capture the conserved changes of receptor or ligand perturbations in cell lines by calculating consensus signatures from all cell lines and perturbation pairs in LINCS L1000. However, some consensus signatures may still have poor quality due to insufficient or ineffective genetic perturbation and drug dosage. We employ receptor filtering based on cross-validation. Briefly, we used OLS to fit linear models for each of the five modalities separately, and then, using the parameter matrices of each modality, we inferred receptor activities from the perturbation signatures of the other modalities with the methods described in sections 4.2 and 4.3.

To assess performance, we calculated the ROC AUC values based on receptor activity vectors and the true value vector, which indicates whether the receptor was perturbed in the sample. Inhibiting and activating permutations were handled separately. First, for inhibition perturbation, all the activation perturbations were not considered (set to 0), and the -1 values were reversed to 1. Then, for activation perturbation, all the receptor inhibitions (-1) were not considered in the true vector. The ROC AUC values were calculated separately, resulting in 2 values for receptors with perturbations in both directions. For negative values, we considered 1-ROC AUC. We kept receptors that are predictive in at least one case with a minimum ROC AUC of 0.6, either in the positive or negative direction. Non-predictive receptors were defined as those where the ROC AUC was below the threshold of 0.6 in both the positive and negative directions. We excluded cases where the CRISPR model predicted shRNA perturbation and conversely because if the receptor is not present in the cell, the receptor knock-out or knock-down will give us incorrect information about the potential gene expression change upon receptor perturbation. Finally, after filtering the non-predictive receptors, we fitted OLS linear models using all the remaining signatures, including all modalities. This resulted in a model containing 229 different receptors.

### 4.5. Assigning confidence levels to receptors

The dataset was randomly split into training and test sets five times, with each set comprising half of the signatures. A model was trained on each training set and then used to predict receptor activity on the corresponding test set. The

Mann-Whitney U (MWU) test was conducted on each split to determine whether there is a significant difference in receptor activities between the perturbed and non-perturbed conditions, based on the predicted receptor activities and the prior knowledge of receptor perturbations. Inhibitory (-1) and stimulatory (1) perturbations were analyzed separately, as was done for ROC AUC calculations in section 4.4. Receptors that had perturbations in both directions had their statistics averaged to obtain a single score for each receptor per split. An overall mean value was calculated for each receptor across splits. ROC AUC values were calculated in the same workflow. Confidence levels for the receptors were assigned based on the aggregated MWU p-values, with confidence levels A-D determined by categorizing p-values into quartiles. Receptors with fewer than 8 signatures in the evaluation were given an E confidence level.

### 4.6. Benchmarking: comparing performance with the CytoSig model

We compared the performance of CytoSig [14] with the RIDDEN model in two settings, where the prediction of the cytokine receptor activities or cytokine signaling activities are evaluated.

To compare the models directly, we assessed the predictive performance using the ground truth dataset of the other model containing perturbation profiles of cytokines and cytokine receptors. First, we inferred the activities of the cytokine receptors using RIDDEN (with the method described in section 4.3), the receptors that have been perturbed by their ligands (cytokines) in the CytoSig dataset. Then, we used the CytoSig model to calculate the cytokine signaling activities from RIDDEN's cytokine and receptor perturbation signatures, which consist of not only the landmark genes but the inferred genes as well to ensure the comparison is equitable. When evaluating RIDDEN on the CytoSig dataset, we mapped receptors to their interacting cytokines based on the ligand-receptor interaction table from OmniPath [21,22], allowing for both one-to-one and one-to-many interactions. When evaluating CytoSig on the RIDDEN dataset, we mapped cytokines to their corresponding receptors while accounting for the type of perturbation applied, which can be inhibitory or activatory. We evaluated the models' predictive performance using the ROC AUC metric (using the scikit-learn Python package), with separate ROC curves for each receptor. Perturbed ligands or receptors were denoted by 1 in the true values vector, while non-perturbed ones were denoted by 0. We handled different perturbation directions separately (similarly as described in section 4.4.), calculating AUC values for stimulatory and 1-AUC values for inhibitory perturbations. The maximum ROC AUC values were selected for a cytokine with both types of perturbations.

As a ground truth dataset, we used the perturbational single-cell RNA-sequencing profiles of Immune Dictionary [15]. After obtaining the cytokine response profiles from the Immune Dictionary's web portal, the perturbation signatures were normalised using gene-wise z-scores for each immune cell type, and then the cell signatures were aggregated to obtain the average cytokine perturbation signatures of the immune cell types. Signatures containing fewer than ten differentially expressed genes (DEG) were filtered out, as they do not capture the translational changes of the perturbation. The number of DEG of the cytokine perturbations was obtained from the study [15].

We inferred the cytokine signaling activities with CytoSig and the cytokine receptor activities using RIDDEN (as described in section 4.3) of the aggregated cytokine perturbation profiles of immune cell types. We assessed the model performances in inferring the perturbed cytokines. We filtered for the overlapping cytokines and their receptors between the CytoSig, RIDDEN and the dataset and evaluated the performance of the models on this subset using the ROC AUC metric, where the perturbed cytokines indicate the ground truth labels. We used prior knowledge of ligand-receptor interactions to map the RIDDEN receptors to their interacting cytokines. We calculated the ROC AUC for RIDDEN receptors with different confidence scores using all possible cytokine receptors whose ligand was perturbed in the dataset.

### 4.7. Benchmarking: NicheNet model ligand activity prediction evaluation on immune dictionary

We used NicheNet [10] to calculate ligand activity scores based on the transcriptomic response of cells to cytokine perturbations from the Immune Dictionary [13,15]. We applied a sender-agnostic approach to evaluate all LR interactions where the cognate receptors are expressed in the receiver cells. For each of the cytokine perturbations in the Immune Dictionary,

we calculated the ligand (cytokine) activities for each single-cell type (acting as the receiver). We excluded signatures with fewer than ten DEGs. We assessed the NicheNet model performance using the ROC AUC metric, with the perturbed cytokines as true labels. We used AUPR (Area Under the Precision-Recall Curve) as a metric for ligand activity, previously suggested as the most informative measure for defining ligand activity [10]. In total, we evaluated 64 cytokine perturbations for which activity can be estimated using NicheNet.

#### 4.8. Evaluation of the model assuming activities of transcription factors can be affected by active receptors

We examined whether the model reflects biological processes according to upstream activated/inhibited receptors that can influence the activity of the downstream transcription factors (TFs). To achieve this, we used the decoupleR [28] Python package to estimate TF activities from the dataset using the TF regulons from DoRothEA [20]. We used KEGG pathways [39] to identify TFs and receptors with shared pathways. For each group of receptors, classified by confidence levels, we compared the activities of TFs included or not in the same KEGG pathways as the receptors. We evaluated the difference by Student's t-test.

We further examined whether receptor–TF pairs within the same KEGG signaling pathway exhibit higher correlations than unrelated pairs, and whether this relationship is better captured by TF activity than by TF gene expression. We used TF expression and TF activity profiles from the CytoSig dataset. of 204 TFs. We computed Pearson correlations between RIDDEN-inferred receptor activities and both (1) TF activity scores [20] and (2) TF gene expression levels. We then compared the distribution of correlation values between the within the KEGG pathway and out-of-pathway groups using Student's t-test.

#### 4.9. Receptor clustering

To investigate how receptors with similar mechanisms of action or different receptor families are represented in our model, we used hierarchical clustering and dendrogram algorithms (from the Scipy Python package [60]) on receptor model coefficients to visualise the similarities between receptor vectors. We obtained receptor family classifications from The IUPHAR/BPS Guide to Pharmacology [61].

#### 4.10. Gene Ontology enrichment based on the RIDDEN's receptor weights

We performed Gene Ontology (GO) Biological Process gene set enrichment analysis (GSEA) on absolute gene weight vectors of the receptors using the decoupleR package [28]. P-values were corrected using the false discovery rate method. We show the significantly enriched terms for the selected receptors with the five highest normalized enrichment scores.

#### 4.11. Biological relevance evaluation through correlation analysis using cell line and patient data

We leveraged patients' gene expression profiles of The Cancer Genome Atlas and cell lines' baseline gene expression profiles of the Cancer Cell Line Encyclopedia to demonstrate that the model can predict potentially activated and inhibited receptors in a large cohort of untreated samples. We inferred receptor activities from the gene expression of the samples (Methods 4.3.) We calculated random distribution by gene label permutation performed 100 times. Subsequently, we calculated Pearson's correlation (with the Scipy Python package [60]) between receptor transcript per million (TPM) values multiplied by ligand TPM values and the inferred receptor activities. We compared the random with the original receptor activity distribution using the Wilcoxon test.

#### 4.12. Receptor activity and patient survival association using pretreated samples

We leveraged gene expression of pretreatment samples and the overall survival of patients from publicly available data [43]. We inferred receptor activities. Then, we investigated the relationship between the activities and the patient response

to therapy using the log-rank test and the Cox regression analysis (with the lifelines Python package [62]). In the log-rank test, we separated patient groups by the mean of the gene expression or receptor activity. In the Cox regression, we used the following equation: $h(t|x) = h_0(t) * \exp(\beta * X)$, where $h(t|x)$ denotes the hazard function, $h_0(t)$ is a baseline hazard, $\beta$ is the coefficient and X denotes the receptor activity or the gene expression.

### 4.13. Receptor activity in patients' single-cell transcriptomic profiles

To show the cell-wise resolution of receptor activity, we used single-cell transcriptomic profiles of tumor patients' samples from a recent publication [44]. We used only the patients' data with clear cell renal cell carcinoma (ccRCC). Analysis was performed by Scanpy [63]. First, cells containing fewer than 300 genes expressed and genes expressed in fewer than 5 cells were removed from the dataset. Then, potential cell doublets identified by scrublet [64], as well as mitochondrial, ribosomal, and sex genes were removed. Cells with more than 25% mitochondrial content were excluded, and cells where the percentage of ribosomal gene counts exceeded 5% were kept. The total number of counts was normalised to 5000 per cell. Then, the data was log-transformed. Highly variable genes were identified, and batch correction was performed based on highly variable genes using the BBKNN algorithm [65]. Receptor activities were calculated on log-normalized counts before batch correction. We visualised the cells using UMAP 2-D projection. We used the cell line annotations provided in the publication, with an extension of grouping cells that contain different marker genes but are classified into the same cell type, like different endothelial cells.

### Supporting information

**S1 Fig. Distribution of z-scored expression values for all landmark genes across the receptor perturbations.**
(TIFF)

**S2 Fig. NicheNet model performance on Immune Dictionary. Boxplot showing ROC AUC values for ligand activity prediction on the Immune Dictionary across 64 cytokine perturbations. The median ROC AUC value (central line and value), first and third quartile (box), minimum and maximum non-outlier values (whiskers) are shown on the boxplot.**
(TIFF)

**S3 Fig. Pearson correlation between receptor and TF activities or TF expression, informed by whether the pairs intersect with the same KEGG pathway or not. Correlations were calculated using absolute values, and the analysis was limited to TFs with measured expression in CytoSig and TF activity profiles. TF–receptor pairs within the same pathway exhibit significantly higher correlations than unrelated pairs, the difference between within-pathway and out-of-pathway correlations is greater when using TF activity compared to TF gene expression.**
(TIFF)

**S4 Fig. Correlation analysis of the CCLE samples receptor * ligand TPM values and their receptor activities. The distribution of correlation values in the dataset (original. left) and the random distribution (1000 gene label permutation. right) are represented in a violin plot. A dashed line indicates the quartiles of the correlation values.**
(TIFF)

**S1 Table. Table of Gene Ontology enrichment analysis for five receptors (CXCR4, IFNGR1, IL6R, TNFRSF1B, TGFBR1), based on the absolute gene weights derived from the RIDDEN model. The top five significantly (FDR<0.05) enriched biological process terms are listed for each receptor.**
(CSV)

**S2 Table.  Table of Cox regression analysis results of nivolumab-treated patient overall survival and receptor activities with sample collection maximum 1 year before therapy.**
(CSV)

**S3 Table.  Table of Cox regression analysis results of nivolumab-treated patient overall survival and gene expression of receptor genes with sample collection maximum 1 year before therapy.**
(CSV)

**S4 Table.  Table of Cox regression analysis results of everolimus-treated patient overall survival and receptor activities with sample collection maximum 1 year before therapy.**
(CSV)

**S1 File.  Graphical abstract.**
(TIFF)

## Acknowledgments

The graphical abstract, Figs 1A, B and 2A were created with BioRender.com.

## Author contributions

**Conceptualization:** Szilvia Barsi, Bence Szalai.

**Data curation:** Szilvia Barsi, Eszter Varga.

**Formal analysis:** Szilvia Barsi, Eszter Varga.

**Funding acquisition:** László Hunyady, Bence Szalai.

**Investigation:** Szilvia Barsi.

**Methodology:** Szilvia Barsi, Daniel Dimitrov.

**Project administration:** Bence Szalai.

**Software:** Szilvia Barsi.

**Supervision:** Julio Saez-Rodriguez, László Hunyady, Bence Szalai.

**Visualization:** Szilvia Barsi.

**Writing – original draft:** Szilvia Barsi.

**Writing – review & editing:** Szilvia Barsi, Eszter Varga, Daniel Dimitrov, Julio Saez-Rodriguez, László Hunyady, Bence Szalai.

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
