## [Decision Letter · Decision Letter 0]

PCOMPBIOL-D-24-02137

RIDDEN: Data-driven inference of receptor activity from transcriptomic data

PLOS Computational Biology

Dear Dr. Szalai,

Thank you for submitting your manuscript to PLOS Computational Biology. After careful consideration, we feel that it has merit but does not fully meet PLOS Computational Biology's publication criteria as it currently stands. Therefore, we invite you to submit a revised version of the manuscript that addresses the points raised during the review process.

Please submit your revised manuscript within 60 days Mar 22 2025 11:59PM. If you will need more time than this to complete your revisions, please reply to this message or contact the journal office at ploscompbiol@plos.org. Please include the following items when submitting your revised manuscript:

We look forward to receiving your revised manuscript.

Kind regards,

Joshua Welch

Academic Editor

PLOS Computational Biology

Pedro Mendes

Section Editor

PLOS Computational Biology

**Additional Editor Comments :**

Please address all reviewer comments, particularly the suggestions for modifying the method itself and additional benchmarking.

**Journal Requirements:**

At this stage, the following Authors/Authors require contributions: Szilvia Barsi, Eszter Varga, Daniel Dimitrov, Julio Saez-Rodriguez, László Hunyady, and Bence Szalai. Please ensure that the full contributions of each author are acknowledged in the "Add/Edit/Remove Authors" section of our submission form.

3) Please upload a copy of Figures 3D, 3E, and 3F which you refer to in your text on page 19 lines 356 and 367. Or, if the figures are no longer to be included as part of the submission please remove all reference to them within the text.

4) Please ensure that all Figure files have corresponding citations and legends within the manuscript. Currently, Figure 4 in your submission file inventory does not have an in-text citation. If the figure is no longer to be included as part of the submission, please remove it from the file inventory.

5) We notice that your supplementary Figure is included in the manuscript file. Please remove it and upload it with the file type 'Supporting Information'. Please ensure that each Supporting Information file has a legend listed in the manuscript after the references list.

1) State what role the funders took in the study. If the funders had no role in your study, please state: "The funders had no role in study design, data collection and analysis, decision to publish, or preparation of the manuscript.".

7) Thank you for stating that “The graphical abstract, Fig 1A, B and Fig 2A figures were created with BioRender.com”. Please include “ created with Biorender.com” in the figure legends.

**Reviewers' comments:**

Reviewer's Responses to Questions

**Comments to the Authors:**

**Please note that one of the reviews is uploaded as an attachment.**

Reviewer #1: Thank you for your great effort on this work. Personally I really appreciate the novel application of RIDDEN on cancer therapy. However, I have a few comments on the methodology part.

First, why you chose Ordinary Least Squares (OLS) linear regression model to infer the association between receptors and other genes? OLS requires the error (residual) term follows normal distribution. Could you prove that? I know you could normalize the gene expression data to make them approximate to normal distribution, but the distribution of "y" in the regression model needs to be shown.

Second, have you ever tried negative binomial regression model? Since negative binomial distribution is that best one to characterize gene expression profiles of transcriptomic data, can you try this one and check whether you get a better result? To construct a valid statistical model, it is essential to analyze the data distribution and select a model that accurately reflects it.

Overall, this is a great and useful work. I recommend this work accepted after solving my comments above. Thanks.

Reviewer #2: The manuscript proposes a computational method, RIDDEN, to predict receptor activities from bulk and single-cell transcriptomics data. The assumption made in this manuscript is that the receptor activities are positively correlated with the target gene expression, which distinguishes this manuscript from the related cell-cell interaction prediction methods that focus on ligand and receptor gene expression. RIDDEN uses publicly available perturbation data to infer the target genes corresponding to each receptor, and use the inferred target gene expression to predict receptor activities of a new data. Despite the novelty of the idea, I have the following concerns.

Major comments

1. Even though the development of RIDDEN is motivated by that previous cell-cell interaction methods do not focus on the receptor side, RIDDEN is never compared against the existing cell-cell interaction inference methods based on both ligand expression and receptor expression. For example, the existing cell-cell interaction prediction methods can be applied to the Immune Dictionary dataset to predict whether each ligand-receptor pair is interacting, which can further predict whether a ligand/receptor is perturbed or not by checking whether it is interacting with any other receptors/ligands. This will allow a direct comparison with RIDDEN and CytoSig.

2. In Fig 2C, “RIDDEN ALL overlap” and “CytoSig overlap” seem to have different number of points in the barplot. Why are they different on the “overlap” set?

3. AUROC value is sensitive to the class imbalance, and therefore the values of AUROC on different datasets may not be directly comparable. In the comparison between RIDDEN and CytoSig on each other’s training datasets, I wonder what’s the class imbalance of each training datasets (percentages of label 0s and label 1s)? And if sub-sample each dataset to have the same class balance, what’s the AUROC values are?

4. When evaluating RIDDEN and CytoSig on each other’s dataset, mapping cytokines to their receptors, and mapping receptors to the binding cytokines are used. I wonder how many cytokines/receptors have one-to-one mapping? If they don’t have a one-to-one-mapping, how is the mapping handled exactly?

5. What dataset is used for section 2.3 “Model evaluation and correlation with biology”?

6. In the analysis of TF expression and predicted receptor activities, I can think two other ways to strengthen the agreement between TF expression and RIDDEN prediction, besides what’s shown in Fig 3B. (1) Whether TFs from the same pathway has a high weight in the RIDDEN receptor-gene weight matrix. (2) For each receptor and each TF in the same pathway, whether the cells/samples with low receptor activities and cells/samples with high receptor activity have differential gene expression of the TF. I suggest also include these analyses or providing a justification of why only consider the analysis in Fig 3B.

7. Regarding the methods.

(1). Ordinary least square is used to infer the receptor-gene weight matrix without regularization, which is the main matrix in RIDDEN and represents the effect on target genes of each receptor. However, OLS typically infers a dense coefficient vector, where almost all coefficients are nonzero, which indicate all receptors will have an effect on all genes. This doesn’t seem correct biologically. In addition, dense coefficients may lead to overfitting to the training data and reduced generalization performance on new datasets. I suggest either trying a few regularization schemes or providing a justification of using the dense coefficient vector in OLS.

(2). The training data includes the perturbation of ligands and mapping the ligands to their corresponding receptors in the design matrix in OLS. However, ligand expression doesn’t have a high correlation with the receptor expression nor receptor activities, which is the motivation of RIDDEN method and supported by result Fig 3C. I wonder whether including the ligand perturbation is necessary or not, if excluding these perturbations, what is the accuracy of RIDDEN?

(3). Does RIDDEN require a minimum number of perturbation experiments for each receptor?

(4). Section 4.1 uses the term “signature” for LINCS database, and this signature seems to have a different meaning than the gene expression signatures per receptor inferred by RIDDEN. What does the “signature” mean here? And why “the receptor and ligand perturbation signatures” are filtered out instead of being retained?

Minor comments

1. In Fig 1A, a transposition is needed of RIDDEN matrix to make the dimensions match in the matrix multiplication.

2. The results in Fig 1C, 2BC, and 3B need clarification.

(1). Fig 1C indicates that the categorization of receptors into confidence levels can be performed using different types of perturbation experiments, and it’s unclear how consistent the results are across them.

(2). It’s unclear which categorization (which x axis in Fig 1C) is used to generate Fig 2BC and Fig 3B.

Reviewer #3: Please see the attached review. In short, I recommend accepting this work after minor revisions.

**Have the authors made all data and (if applicable) computational code underlying the findings in their manuscript fully available?**

Reviewer #1: **No: ** The authors have published all code and the receptor-gene weight matrix, but they do not publish the data they used to train the linear model. I thought this is important to see how they collected and processed the datasets.

Reviewer #2: None

Reviewer #3: Yes

PLOS authors have the option to publish the peer review history of their article (what does this mean? ). If published, this will include your full peer review and any attached files.

**Do you want your identity to be public for this peer review?** For information about this choice, including consent withdrawal, please see our Privacy Policy .

Reviewer #1: No

Reviewer #2: No

Reviewer #3: No

**Figure resubmission:**
---

## [Decision Letter · Decision Letter 1]

PCOMPBIOL-D-24-02137R1

RIDDEN: Data-driven inference of receptor activity from transcriptomic data

PLOS Computational Biology

Dear Dr. Szalai,

Thank you for submitting your manuscript to PLOS Computational Biology. After careful consideration, we feel that it has merit but does not fully meet PLOS Computational Biology's publication criteria as it currently stands. Therefore, we invite you to submit a revised version of the manuscript that addresses the points raised during the review process.

Please submit your revised manuscript within 30 days Jul 19 2025 11:59PM. If you will need more time than this to complete your revisions, please reply to this message or contact the journal office at ploscompbiol@plos.org. Please include the following items when submitting your revised manuscript:

We look forward to receiving your revised manuscript.

Kind regards,

Joshua Welch

Academic Editor

PLOS Computational Biology

Pedro Mendes

Section Editor

PLOS Computational Biology

**Additional Editor Comments :**

The reviewers indicated that you partially addressed their comments. Please address the remaining reviewer requests, which primarily request clarification in the text.

**Reviewers' comments:**

Reviewer's Responses to Questions

Reviewer #1: The authors addressed most of my questions / concerns. However, there are a few comments left.

1. The authors have provided a justification for the use of OLS regression, but does not add them in the manuscript.

From the figure in the response, the processed data from LINCS seem to follow normal distribution. It would be better if authors can show this model diagnosis plot as a supplementary figure to support the usage of OLS regression model. It would be perfect if hypothesis testing like shapiro test can be used to strictly show the data follows normal distribution.

2. The data used to build the model has been publicly available, but the link is not valid in the 'Data Availability' section.

I found that the data is now publicly available from the Github page of RIDDEN. However, the link provided by the author in 'Data Availability' section does not work.

Minor comments

I suggest the authors consider revising the color scheme of the figures. Currently, most plots are based on blue tones, which may make it difficult to distinguish between elements or reduce visual impact. Incorporating a more diverse or contrasting color palette could improve clarity and readability.

Once these points are addressed, I would be happy to recommend acceptance.

Reviewer #2: The authors addressed most of my previous comments, while I still want to follow up a few points.

1. Regarding model evaluation using the relationship between receptor activity and TF activity, and RIDDEN model assumption between receptor activity and downstream gene expression change. It seems the assumption is that receptor activity is correlated with the downstream gene expression change, as well as TF “activity” change, but not TF gene expression change. Is that right? I suggest add a few sentences clearly state the assumption with justification.

2. Regarding OLS with regularization, it’s good to know that LASSO regularization doesn’t change the performance of RIDDEN by a lot. But I think it’s still important to directly ensure RIDDEN weight matrix is biologically meaningful. I suggest looking into enriched pathways of up and down regulated genes of a few receptors based on RIDDEN weight matrix, and verify in previous literature whether the receptor is known to be related to the enriched pathways. As RIDDEN is a linear model, the high positive weight can be directly interpreted as up-regulation of the gene by receptor activation, vice versa for low negative weights. This evaluation also sheds light on the lack of information about the regulatory interactions for other signaling proteins than transcription factors, which was used as motivation in the intro.

3. Regarding whether RIDDEN matrix should be transposed in Fig 1 or not. Fig 1B is correct, but in Fig 1A the matrix multiplication currently writes as perturbation profile (cells-by-genes) = design matrix (cells-by-receptors) * RIDDEN matrix (genes-by-receptors), which is incorrect in dimensions.

Reviewer #3: I am satisfied with the response to reviewer questions and comments and think the authors have done enough for the manuscript to be published.

**Have the authors made all data and (if applicable) computational code underlying the findings in their manuscript fully available?**

Reviewer #1: Yes

Reviewer #2: Yes

Reviewer #3: Yes

PLOS authors have the option to publish the peer review history of their article (what does this mean? ). If published, this will include your full peer review and any attached files.

**Do you want your identity to be public for this peer review?** For information about this choice, including consent withdrawal, please see our Privacy Policy .

Reviewer #1: No

Reviewer #2: No

Reviewer #3: No

**Figure resubmission:**
---

## [Editor Report · Decision Letter 2]

Dear Dr. Szalai,

We are pleased to inform you that your manuscript 'RIDDEN: Data-driven inference of receptor activity from transcriptomic data' has been provisionally accepted for publication in PLOS Computational Biology.

Best regards,

Joshua Welch

Academic Editor

PLOS Computational Biology

Pedro Mendes

Section Editor

PLOS Computational Biology

---

## [Editor Report · Acceptance letter]

PCOMPBIOL-D-24-02137R2

RIDDEN: Data-driven inference of receptor activity from transcriptomic data

Dear Dr Szalai,

I am pleased to inform you that your manuscript has been formally accepted for publication in PLOS Computational Biology. Your manuscript is now with our production department and you will be notified of the publication date in due course.

With kind regards,

Zsofia Freund
